# Ranking via Robust Binary Classification

**Hyokun Yun**
Amazon
Seattle, WA 98109
yunhyoku@amazon.com

**Parameswaran Raman, S. V. N. Vishwanathan**
Department of Computer Science
University of California
Santa Cruz, CA 95064
{params,vishy}@ucsc.edu

## Abstract

We propose RoBiRank, a ranking algorithm that is motivated by observing a close connection between evaluation metrics for learning to rank and loss functions for robust classification. It shows competitive performance on standard benchmark datasets against a number of other representative algorithms in the literature. We also discuss extensions of RoBiRank to large scale problems where explicit feature vectors and scores are not given. We show that RoBiRank can be efficiently parallelized across a large number of machines; for a task that requires $386,133 \times 49,824,519$ pairwise interactions between items to be ranked, RoBiRank finds solutions that are of dramatically higher quality than that can be found by a state-of-the-art competitor algorithm, given the same amount of wall-clock time for computation.

## 1 Introduction

Learning to rank is a problem of ordering a set of items according to their relevances to a given context [8]. While a number of approaches have been proposed in the literature, in this paper we provide a new perspective by showing a close connection between ranking and a seemingly unrelated topic in machine learning, namely, robust binary classification.

In robust classification [13], we are asked to learn a classifier in the presence of outliers. Standard models for classification such as Support Vector Machines (SVMs) and logistic regression do not perform well in this setting, since the convexity of their loss functions does not let them give up their performance on any of the data points [16]; for a classification model to be robust to outliers, it has to be capable of sacrificing its performance on some of the data points. We observe that this requirement is very similar to what standard metrics for ranking try to evaluate. Discounted Cumulative Gain (DCG) [17] and its normalized version NDCG, popular metrics for learning to rank, strongly emphasize the performance of a ranking algorithm at the top of the list; therefore, a good ranking algorithm in terms of these metrics has to be able to give up its performance at the bottom of the list if that can improve its performance at the top.

In fact, we will show that DCG and NDCG can indeed be written as a natural generalization of robust loss functions for binary classification. Based on this observation we formulate RoBiRank, a novel model for ranking, which maximizes the lower bound of (N)DCG. Although the non-convexity seems unavoidable for the bound to be tight [9], our bound is based on the class of robust loss functions that are found to be empirically easier to optimize [10]. Indeed, our experimental results suggest that RoBiRank reliably converges to a solution that is competitive as compared to other representative algorithms even though its objective function is non-convex.

While standard deterministic optimization algorithms such as L-BFGS [19] can be used to estimate parameters of RoBiRank, to apply the model to large-scale datasets a more efficient parameter estimation algorithm is necessary. This is of particular interest in the context of latent collaborative

retrieval [24]; unlike standard ranking task, here the number of items to rank is very large and explicit feature vectors and scores are not given.

Therefore, we develop an efficient parallel stochastic optimization algorithm for this problem. It has two very attractive characteristics: First, the time complexity of each stochastic update is independent of the size of the dataset. Also, when the algorithm is distributed across multiple number of machines, no interaction between machines is required during most part of the execution; therefore, the algorithm enjoys near linear scaling. This is a significant advantage over serial algorithms, since it is very easy to deploy a large number of machines nowadays thanks to the popularity of cloud computing services, e.g. Amazon Web Services.

We apply our algorithm to latent collaborative retrieval task on Million Song Dataset [3] which consists of 1,129,318 users, 386,133 songs, and 49,824,519 records; for this task, a ranking algorithm has to optimize an objective function that consists of $386,133 \times 49,824,519$ number of pairwise interactions. With the same amount of wall-clock time given to each algorithm, RoBiRank leverages parallel computing to outperform the state-of-the-art with a 100% lift on the evaluation metric.

## 2 Robust Binary Classification

Suppose we are given training data which consists of $n$ data points $(x_1, y_1), (x_2, y_2), \ldots, (x_n, y_n)$, where each $x_i \in \mathbb{R}^d$ is a $d$-dimensional feature vector and $y_i \in \{-1, +1\}$ is a label associated with it. A linear model attempts to learn a $d$-dimensional parameter $\omega$, and for a given feature vector $x$ it predicts label $+1$ if $\langle x, \omega \rangle \geq 0$ and $-1$ otherwise. Here $\langle \cdot, \cdot \rangle$ denotes the Euclidean dot product between two vectors. The quality of $\omega$ can be measured by the number of mistakes it makes: $L(\omega) := \sum_{i=1}^{n} I(y_i \cdot \langle x_i, \omega \rangle < 0)$. The indicator function $I(\cdot < 0)$ is called the 0-1 loss function, because it has a value of 1 if the decision rule makes a mistake, and 0 otherwise. Unfortunately, since the 0-1 loss is a discrete function its minimization is difficult [11]. The most popular solution to this problem in machine learning is to upper bound the 0-1 loss by an easy to optimize function [2]. For example, logistic regression uses the logistic loss function $\sigma_0(t) := \log_2(1 + 2^{-t})$, to come up with a continuous and convex objective function

$$\overline{L}(\omega) := \sum_{i=1}^{n} \sigma_0(y_i \cdot \langle x_i, \omega \rangle), \tag{1}$$

which upper bounds $L(\omega)$. It is clear that for each $i$, $\sigma_0(y_i \cdot \langle x_i, \omega \rangle)$ is a convex function in $\omega$; therefore, $\overline{L}(\omega)$, a sum of convex functions, is also a convex function which is relatively easier to optimize [6]. Support Vector Machines (SVMs) on the other hand can be recovered by using the hinge loss to upper bound the 0-1 loss.

However, convex upper bounds such as $\overline{L}(\omega)$ are known to be sensitive to outliers [16]. The basic intuition here is that when $y_i \cdot \langle x_i, \omega \rangle$ is a very large negative number for some data point $i$, $\sigma(y_i \cdot \langle x_i, \omega \rangle)$ is also very large, and therefore the optimal solution of (1) will try to decrease the loss on such outliers at the expense of its performance on "normal" data points.

In order to construct robust loss functions, consider the following two transformation functions:

$$\rho_1(t) := \log_2(t + 1), \quad \rho_2(t) := 1 - \frac{1}{\log_2(t + 2)}, \tag{2}$$

which, in turn, can be used to define the following loss functions:

$$\sigma_1(t) := \rho_1(\sigma_0(t)), \quad \sigma_2(t) := \rho_2(\sigma_0(t)). \tag{3}$$

One can see that $\sigma_1(t) \to \infty$ as $t \to -\infty$, but at a much slower rate than $\sigma_0(t)$ does; its derivative $\sigma_1'(t) \to 0$ as $t \to -\infty$. Therefore, $\sigma_1(\cdot)$ does not grow as rapidly as $\sigma_0(t)$ on hard-to-classify data points. Such loss functions are called Type-I robust loss functions by Ding [10], who also showed that they enjoy statistical robustness properties. $\sigma_2(t)$ behaves even better: $\sigma_2(t)$ converges to a constant as $t \to -\infty$, and therefore "gives up" on hard to classify data points. Such loss functions are called Type-II loss functions, and they also enjoy statistical robustness properties [10].

In terms of computation, of course, $\sigma_1(\cdot)$ and $\sigma_2(\cdot)$ are not convex, and therefore the objective function based on such loss functions is more difficult to optimize. However, it has been observed

in Ding [10] that models based on optimization of Type-I functions are often empirically much more successful than those which optimize Type-II functions. Furthermore, the solutions of Type-I optimization are more stable to the choice of parameter initialization. Intuitively, this is because Type-II functions asymptote to a constant, reducing the gradient to almost zero in a large fraction of the parameter space; therefore, it is difficult for a gradient-based algorithm to determine which direction to pursue. See Ding [10] for more details.

## 3  Ranking Model via Robust Binary Classification

Let $\mathcal{X} = \{x_1, x_2, \ldots, x_n\}$ be a set of contexts, and $\mathcal{Y} = \{y_1, y_2, \ldots, y_m\}$ be a set of items to be ranked. For example, in movie recommender systems $\mathcal{X}$ is the set of users and $\mathcal{Y}$ is the set of movies. In some problem settings, only a subset of $\mathcal{Y}$ is relevant to a given context $x \in \mathcal{X}$; e.g. in document retrieval systems, only a subset of documents is relevant to a query. Therefore, we define $\mathcal{Y}_x \subset \mathcal{Y}$ to be a set of items relevant to context $x$. Observed data can be described by a set $W := \{W_{xy}\}_{x \in \mathcal{X}, y \in \mathcal{Y}_x}$ where $W_{xy}$ is a real-valued score given to item $y$ in context $x$.

We adopt a standard problem setting used in the literature of learning to rank. For each context $x$ and an item $y \in \mathcal{Y}_x$, we aim to learn a scoring function $f(x, y) : \mathcal{X} \times \mathcal{Y}_x \to \mathbb{R}$ that induces a ranking on the item set $\mathcal{Y}_x$; the higher the score, the more important the associated item is in the given context. To learn such a function, we first extract joint features of $x$ and $y$, which will be denoted by $\phi(x, y)$. Then, we parametrize $f(\cdot, \cdot)$ using a parameter $\omega$, which yields the linear model $f_\omega(x, y) := \langle \phi(x, y), \omega \rangle$, where, as before, $\langle \cdot, \cdot \rangle$ denotes the Euclidean dot product between two vectors. $\omega$ induces a ranking on the set of items $\mathcal{Y}_x$; we define $\mathrm{rank}_\omega(x, y)$ to be the rank of item $y$ in a given context $x$ induced by $\omega$. Observe that $\mathrm{rank}_\omega(x, y)$ can also be written as a sum of 0-1 loss functions (see e.g. Usunier et al. [23]):

$$\mathrm{rank}_\omega(x, y) = \sum_{y' \in \mathcal{Y}_x, y' \neq y} I\left(f_\omega(x, y) - f_\omega(x, y') < 0\right). \tag{4}$$

### 3.1  Basic Model

If an item $y$ is very relevant in context $x$, a good parameter $\omega$ should position $y$ at the top of the list; in other words, $\mathrm{rank}_\omega(x, y)$ has to be small, which motivates the following objective function [7]:

$$L(\omega) := \sum_{x \in \mathcal{X}} c_x \sum_{y \in \mathcal{Y}_x} v(W_{xy}) \cdot \mathrm{rank}_\omega(x, y), \tag{5}$$

where $c_x$ is an weighting factor for each context $x$, and $v(\cdot) : \mathbb{R}^+ \to \mathbb{R}^+$ quantifies the relevance level of $y$ on $x$. Note that $\{c_x\}$ and $v(W_{xy})$ can be chosen to reflect the metric the model is going to be evaluated on (this will be discussed in Section 3.2). Note that (5) can be rewritten using (4) as a sum of indicator functions. Following the strategy in Section 2, one can form an upper bound of (5) by bounding each 0-1 loss function by a logistic loss function:

$$\overline{L}(\omega) := \sum_{x \in \mathcal{X}} c_x \sum_{y \in \mathcal{Y}_x} v(W_{xy}) \cdot \sum_{y' \in \mathcal{Y}_x, y' \neq y} \sigma_0\left(f_\omega(x, y) - f_\omega(x, y')\right). \tag{6}$$

Just like (1), (6) is convex in $\omega$ and hence easy to minimize.

### 3.2  DCG

Although (6) enjoys convexity, it may not be a good objective function for ranking. This is because in most applications of learning to rank, it is more important to do well at the top of the list than at the bottom, as users typically pay attention only to the top few items. Therefore, it is desirable to *give up* performance on the lower part of the list in order to gain quality at the top. This intuition is similar to that of robust classification in Section 2; a stronger connection will be shown below.

Discounted Cumulative Gain (DCG) [17] is one of the most popular metrics for ranking. For each context $x \in \mathcal{X}$, it is defined as:

$$\mathrm{DCG}(\omega) := c_x \sum_{y \in \mathcal{Y}_x} \frac{v(W_{xy})}{\log_2(\mathrm{rank}_\omega(x, y) + 2)}, \tag{7}$$

where $v(t) = 2^t - 1$ and $c_x = 1$. Since $1/\log(t+2)$ decreases quickly and then asymptotes to a constant as $t$ increases, this metric emphasizes the quality of the ranking at the top of the list. Normalized DCG (NDCG) simply normalizes the metric to bound it between 0 and 1 by calculating the maximum achievable DCG value $m_x$ and dividing by it [17].

### 3.3 RoBiRank

Now we formulate RoBiRank, which optimizes the lower bound of metrics for ranking in form (7). Observe that $\max_\omega \mathrm{DCG}(\omega)$ can be rewritten as

$$\min_\omega \sum_{x \in \mathcal{X}} c_x \sum_{y \in \mathcal{Y}_x} v\left(W_{xy}\right) \cdot \left\{ 1 - \frac{1}{\log_2\left(\mathrm{rank}_\omega(x,y)+2\right)} \right\}. \qquad (8)$$

Using (4) and the definition of the transformation function $\rho_2(\cdot)$ in (2), we can rewrite the objective function in (8) as:

$$L_2(\omega) := \sum_{x \in \mathcal{X}} c_x \sum_{y \in \mathcal{Y}_x} v\left(W_{xy}\right) \cdot \rho_2\left( \sum_{y' \in \mathcal{Y}_x, y' \neq y} I\left(f_\omega(x,y) - f_\omega(x,y') < 0\right) \right). \qquad (9)$$

Since $\rho_2(\cdot)$ is a monotonically increasing function, we can bound (9) with a continuous function by bounding each indicator function using the logistic loss:

$$\overline{L}_2(\omega) := \sum_{x \in \mathcal{X}} c_x \sum_{y \in \mathcal{Y}_x} v\left(W_{xy}\right) \cdot \rho_2\left( \sum_{y' \in \mathcal{Y}_x, y' \neq y} \sigma_0\left(f_\omega(x,y) - f_\omega(x,y')\right) \right). \qquad (10)$$

This is reminiscent of the basic model in (6); as we applied the transformation $\rho_2(\cdot)$ on the logistic loss $\sigma_0(\cdot)$ to construct the robust loss $\sigma_2(\cdot)$ in (3), we are again applying the same transformation on (6) to construct a loss function that respects the DCG metric used in ranking. In fact, (10) can be seen as a generalization of robust binary classification by applying the transformation on a *group* of logistic losses instead of a single loss. In both robust classification and ranking, the transformation $\rho_2(\cdot)$ enables models to give up on part of the problem to achieve better overall performance.

As we discussed in Section 2, however, transformation of logistic loss using $\rho_2(\cdot)$ results in Type-II loss function, which is very difficult to optimize. Hence, instead of $\rho_2(\cdot)$ we use an alternative transformation $\rho_1(\cdot)$, which generates Type-I loss function, to define the objective function of RoBiRank:

$$\overline{L}_1(\omega) := \sum_{x \in \mathcal{X}} c_x \sum_{y \in \mathcal{Y}_x} v\left(W_{xy}\right) \cdot \rho_1\left( \sum_{y' \in \mathcal{Y}_x, y' \neq y} \sigma_0\left(f_\omega(x,y) - f_\omega(x,y')\right) \right). \qquad (11)$$

Since $\rho_1(t) \geq \rho_2(t)$ for every $t > 0$, we have $\overline{L}_1(\omega) \geq \overline{L}_2(\omega) \geq L_2(\omega)$ for every $\omega$. Note that $\overline{L}_1(\omega)$ is continuous and twice differentiable. Therefore, standard gradient-based optimization techniques can be applied to minimize it. As is standard, a regularizer on $\omega$ can be added to avoid overfitting; for simplicity, we use the $\ell_2$-norm in our experiments.

### 3.4 Standard Learning to Rank Experiments

We conducted experiments to check the performance of RoBiRank (11) in a standard learning to rank setting, with a small number of labels to rank. We pitch RoBiRank against the following algorithms: RankSVM [15], the ranking algorithm of Le and Smola [14] (called LSRank in the sequel), InfNormPush [22], IRPush [1], and 8 standard ranking algorithms implemented in RankLib[1] namely MART, RankNet, RankBoost, AdaRank, CoordAscent, LambdaMART, ListNet and RandomForests. We use three sources of datasets: LETOR 3.0 [8], LETOR 4.0[2] and YAHOO LTRC [20], which are standard benchmarks for ranking (see Table 2 for summary statistics). Each dataset consists of five folds; we consider the first fold, and use the training, validation, and test splits provided. We train with different values of regularization parameter, and select one with the best NDCG

on the validation dataset. The performance of the model with this parameter on the test dataset is reported. We used implementation of the L-BFGS algorithm provided by the Toolkit for Advanced Optimization (TAO)[3] for estimating the parameter of RoBiRank. For the other algorithms, we either implemented them using our framework or used the implementations provided by the authors.

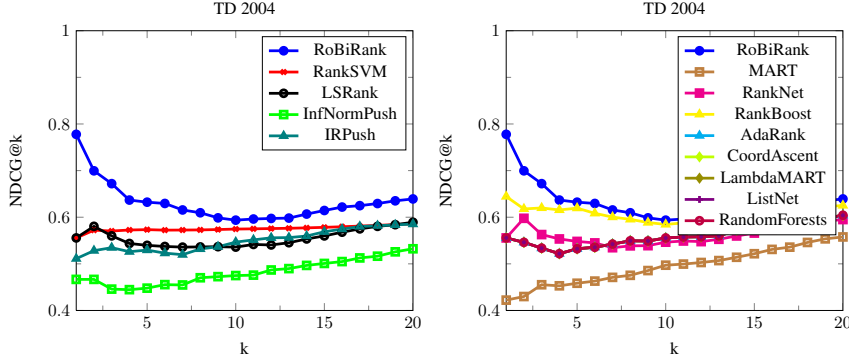

Figure 1: Comparison of RoBiRank with a number of competing algorithms.

We use values of NDCG at different levels of truncation as our evaluation metric [17]; see Figure 1. RoBiRank outperforms its competitors on most of the datasets; due to space constraints, we only present plots for the TD 2004 dataset here and other plots can be found in Appendix B. The performance of RankSVM seems insensitive to the level of truncation for NDCG. On the other hand, RoBiRank, which uses non-convex loss function to concentrate its performance at the top of the ranked list, performs much better especially at low truncation levels. It is also interesting to note that the NDCG@k curve of LSRank is similar to that of RoBiRank, but RoBiRank consistently outperforms at each level. RoBiRank dominates Inf-Push and IR-Push at all levels. When compared to standard algorithms, Figure 1 (right), again RoBiRank outperforms especially at the top of the list.

Overall, RoBiRank outperforms IRPush and InfNormPush on all datasets except TD 2003 and OHSUMED where IRPush seems to fare better at the top of the list. Compared to the 8 standard algorithms, again RobiRank either outperforms or performs comparably to the best algorithm except on two datasets (TD 2003 and HP 2003), where MART and Random Forests overtake RobiRank at few values of NDCG. We present a summary of the NDCG values obtained by each algorithm in Table 2 in the appendix. On the MSLR30K dataset, some of the additional algorithms like InfNorm-Push and IRPush did not complete within the time period available; indicated by dashes in the table.

## 4   Latent Collaborative Retrieval

For each context $x$ and an item $y \in \mathcal{Y}$, the standard problem setting of learning to rank requires training data to contain feature vector $\phi(x, y)$ and score $W_{xy}$ assigned on the $x, y$ pair. When the number of contexts $|\mathcal{X}|$ or the number of items $|\mathcal{Y}|$ is large, it might be difficult to define $\phi(x, y)$ and measure $W_{xy}$ for all $x, y$ pairs. Therefore, in most learning to rank problems we define the set of *relevant* items $\mathcal{Y}_x \subset \mathcal{Y}$ to be much smaller than $\mathcal{Y}$ for each context $x$, and then collect data only for $\mathcal{Y}_x$. Nonetheless, this may not be realistic in all situations; in a movie recommender system, for example, for each user *every* movie is somewhat relevant.

On the other hand, implicit user feedback data is much more abundant. For example, a lot of users on Netflix would simply watch movie streams on the system but do not leave an explicit rating. By the action of watching a movie, however, they implicitly express their preference. Such data consist only of positive feedback, unlike traditional learning to rank datasets which have score $W_{xy}$ between each context-item pair $x, y$. Again, we may not be able to extract feature vectors for each $x, y$ pair.

In such a situation, we can attempt to learn the score function $f(x, y)$ without a feature vector $\phi(x, y)$ by embedding each context and item in an Euclidean latent space; specifically, we redefine the score function to be: $f(x, y) := \langle U_x, V_y \rangle$, where $U_x \in \mathbb{R}^d$ is the embedding of the context $x$ and $V_y \in \mathbb{R}^d$

is that of the item $y$. Then, we can learn these embeddings by a ranking model. This approach was introduced in Weston et al. [24], and was called *latent collaborative retrieval*.

Now we specialize RoBiRank model for this task. Let us define $\Omega$ to be the set of context-item pairs $(x, y)$ which was observed in the dataset. Let $v(W_{xy}) = 1$ if $(x, y) \in \Omega$, and $0$ otherwise; this is a natural choice since the score information is not available. For simplicity, we set $c_x = 1$ for every $x$. Now RoBiRank (11) specializes to:

$$\overline{L}_1(U, V) = \sum_{(x,y) \in \Omega} \rho_1 \left( \sum_{y' \neq y} \sigma_0(f(U_x, V_y) - f(U_x, V_{y'})) \right). \tag{12}$$

Note that now the summation inside the parenthesis of (12) is over all items $\mathcal{Y}$ instead of a smaller set $\mathcal{Y}_x$, therefore we omit specifying the range of $y'$ from now on. To avoid overfitting, a regularizer is added to (12); for simplicity we use the Frobenius norm of $U$ and $V$ in our experiments.

### 4.1 Stochastic Optimization

When the size of the data $|\Omega|$ or the number of items $|\mathcal{Y}|$ is large, however, methods that require exact evaluation of the function value and its gradient will become very slow since the evaluation takes $O(|\Omega| \cdot |\mathcal{Y}|)$ computation. In this case, stochastic optimization methods are desirable [4]; in this subsection, we will develop a stochastic gradient descent algorithm whose complexity is independent of $|\Omega|$ and $|\mathcal{Y}|$.

For simplicity, let $\theta$ be a concatenation of all parameters $\{U_x\}_{x \in \mathcal{X}}$, $\{V_y\}_{y \in \mathcal{Y}}$. The gradient $\nabla_\theta L_1(U, V)$ of (12) is

$$\sum_{(x,y) \in \Omega} \nabla_\theta \rho_1 \left( \sum_{y' \neq y} \sigma_0(f(U_x, V_y) - f(U_x, V_{y'})) \right).$$

Finding an unbiased estimator of the gradient whose computation is independent of $|\Omega|$ is not difficult; if we sample a pair $(x, y)$ uniformly from $\Omega$, then it is easy to see that the following estimator

$$|\Omega| \cdot \nabla_\theta \rho_1 \left( \sum_{y' \neq y} \sigma_0(f(U_x, V_y) - f(U_x, V_{y'})) \right) \tag{13}$$

is unbiased. This still involves a summation over $\mathcal{Y}$, however, so it requires $O(|\mathcal{Y}|)$ calculation. Since $\rho_1(\cdot)$ is a nonlinear function it seems unlikely that an unbiased stochastic gradient which randomizes over $\mathcal{Y}$ can be found; nonetheless, to achieve convergence guarantees of the stochastic gradient descent algorithm, unbiasedness of the estimator is necessary [18].

We attack this problem by *linearizing* the objective function by parameter expansion. Note the following property of $\rho_1(\cdot)$ [5]:

$$\rho_1(t) = \log_2(t+1) \leq -\log_2 \xi + \frac{\xi \cdot (t+1) - 1}{\log 2}. \tag{14}$$

This holds for any $\xi > 0$, and the bound is tight when $\xi = \frac{1}{t+1}$. Now introducing an auxiliary parameter $\xi_{xy}$ for each $(x, y) \in \Omega$ and applying this bound, we obtain an upper bound of (12) as

$$L(U, V, \xi) := \sum_{(x,y) \in \Omega} -\log_2 \xi_{xy} + \frac{\xi_{xy} \left( \sum_{y' \neq y} \sigma_0(f(U_x, V_y) - f(U_x, V_{y'})) + 1 \right) - 1}{\log 2}. \tag{15}$$

Now we propose an iterative algorithm in which, each iteration consists of $(U, V)$-step and $\xi$-step; in the $(U, V)$-step we minimize (15) in $(U, V)$ and in the $\xi$-step we minimize in $\xi$. Pseudo-code can be found in Algorithm 1 in Appendix C.

$(U, V)$**-step** The partial derivative of (15) in terms of $U$ and $V$ can be calculated as: $\nabla_{U,V} L(U, V, \xi) := \frac{1}{\log 2} \sum_{(x,y) \in \Omega} \xi_{xy} \left( \sum_{y' \neq y} \nabla_{U,V} \sigma_0(f(U_x, V_y) - f(U_x, V_{y'})) \right)$. Now it is easy to see that the following stochastic procedure unbiasedly estimates the above gradient:

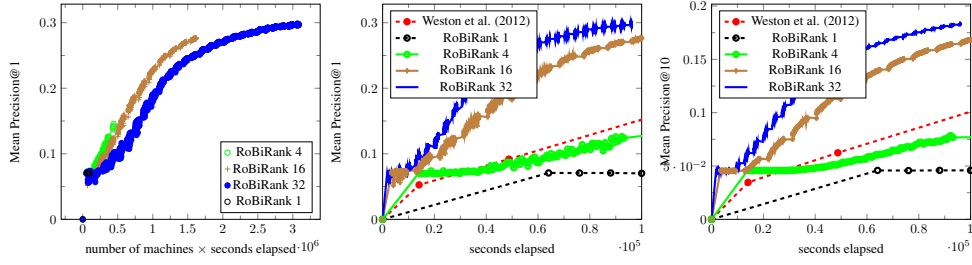

Figure 2: Left: Scaling of RoBiRank on Million Song Dataset. Center, Right: Comparison of RoBiRank and Weston et al. [24] when the same amount of wall-clock computation time is given.

- Sample $(x, y)$ uniformly from $\Omega$
- Sample $y'$ uniformly from $\mathcal{Y} \setminus \{y\}$
- Estimate the gradient by

$$\frac{|\Omega| \cdot (|\mathcal{Y}| - 1) \cdot \xi_{xy}}{\log 2} \cdot \nabla_{U,V} \sigma_0(f(U_x, V_y) - f(U_x, V_{y'})). \tag{16}$$

Therefore a stochastic gradient descent algorithm based on (16) will converge to a local minimum of the objective function (15) with probability one [21]. Note that the time complexity of calculating (16) is independent of $|\Omega|$ and $|\mathcal{Y}|$. Also, it is a function of only $U_x$ and $V_y$; the gradient is zero in terms of other variables.

**$\xi$-step** When $U$ and $V$ are fixed, minimization of $\xi_{xy}$ variable is independent of each other and a simple analytic solution exists: $\xi_{xy} = \frac{1}{\sum_{y' \neq y} \sigma_0(f(U_x, V_y) - f(U_x, V_{y'})) + 1}$. This of course requires $O(|\mathcal{Y}|)$ work. In principle, we can avoid summation over $\mathcal{Y}$ by taking stochastic gradient in terms of $\xi_{xy}$ as we did for $U$ and $V$. However, since the exact solution is simple to compute and also because most of the computation time is spent on $(U, V)$-step, we found this update rule to be efficient.

**Parallelization** The linearization trick in (15) not only enables us to construct an efficient stochastic gradient algorithm, but also makes possible to efficiently parallelize the algorithm across multiple number of machines. Due to lack of space, details are relegated to Appendix D.

## 4.2 Experiments

In this subsection we use the Million Song Dataset (MSD) [3], which consists of 1,129,318 users ($|\mathcal{X}|$), 386,133 songs ($|\mathcal{Y}|$), and 49,824,519 records ($|\Omega|$) of a user $x$ playing a song $y$ in the training dataset. The objective is to predict the songs from the test dataset that a user is going to listen to[4].

Since explicit ratings are not given, NDCG is not applicable for this task; we use precision at 1 and 10 [17] as our evaluation metric. In our first experiment we study the scaling behavior of RoBiRank as a function of number of machines. RoBiRank $p$ denotes the parallel version of RoBiRank which is distributed across $p$ machines. In Figure 2 (left) we plot mean Precision@1 as a function of the number of machines $\times$ the number of seconds elapsed; this is a proxy for CPU time. If an algorithm linearly scales across multiple processors, then all lines in the figure should overlap with each other. As can be seen RoBiRank exhibits near ideal speed up when going from 4 to 32 machines[5].

In our next experiment we compare RoBiRank with a state of the art algorithm from Weston et al. [24], which optimizes a similar objective function (17). We compare how fast the quality of the solution improves as a function of wall clock time. Since the authors of Weston et al. [24] do not make available their code, we implemented their algorithm within our framework using the same data structures and libraries used by our method. Furthermore, for a fair comparison, we used the same initialization for $U$ and $V$ and performed an identical grid-search over the step size parameter.

It can be seen from Figure 2 (center, right) that on a single machine the algorithm of Weston et al. [24] is very competitive and outperforms RoBiRank. The reason for this might be the introduction of the additional $\xi$ variables in RoBiRank, which slows down convergence. However, RoBiRank training can be distributed across processors, while it is not clear how to parallelize the algorithm of Weston et al. [24]. Consequently, RoBiRank 32 which uses 32 machines for its computation can produce a significantly better model within the same wall clock time window.

## 5   Related Work

In terms of modeling, viewing ranking problems as generalization of binary classification problems is not a new idea; for example, RankSVM defines the objective function as a sum of hinge losses, similarly to our basic model (6) in Section 3.1. However, it does not directly optimize the ranking metric such as NDCG; the objective function and the metric are not immediately related to each other. In this respect, our approach is closer to that of Le and Smola [14] which constructs a convex upper bound on the ranking metric and Chapelle et al. [9] which improves the bound by introducing non-convexity. The objective function of Chapelle et al. [9] is also motivated by ramp loss, which is used for robust classification; nonetheless, to our knowledge the direct connection between the ranking metrics in form (7) (DCG, NDCG) and the robust loss (3) is our novel contribution. Also, our objective function is designed to specifically bound the ranking metric, while Chapelle et al. [9] proposes a general recipe to improve existing convex bounds.

Stochastic optimization of the objective function for latent collaborative retrieval has been also explored in Weston et al. [24]. They attempt to minimize

$$\sum_{(x,y)\in\Omega} \Phi\left(1 + \sum_{y'\neq y} I(f(U_x, V_y) - f(U_x, V_{y'}) < 0)\right),\tag{17}$$

where $\Phi(t) = \sum_{k=1}^{t} \frac{1}{k}$. This is similar to our objective function (15); $\Phi(\cdot)$ and $\rho_2(\cdot)$ are asymptotically equivalent. However, we argue that our formulation (15) has two major advantages. First, it is a continuous and differentiable function, therefore gradient-based algorithms such as L-BFGS and stochastic gradient descent have convergence guarantees. On the other hand, the objective function of Weston et al. [24] is not even continuous, since their formulation is based on a function $\Phi(\cdot)$ that is defined for only natural numbers. Also, through the linearization trick in (15) we are able to obtain an unbiased stochastic gradient, which is necessary for the convergence guarantee, and to parallelize the algorithm across multiple machines as discussed in Appendix D. It is unclear how these techniques can be adapted for the objective function of Weston et al. [24].

## 6   Conclusion

In this paper, we developed RoBiRank, a novel model on ranking, based on insights and techniques from robust binary classification. Then, we proposed a scalable and parallelizable stochastic optimization algorithm that can be applied to latent collaborative retrieval task which large-scale data without feature vectors and explicit scores have to take care of. Experimental results on both learning to rank datasets and latent collaborative retrieval dataset suggest the advantage of our approach.

As a final note, the experiments in Section 4.2 are arguably unfair towards WSABIE. For instance, one could envisage using clever engineering tricks to derive a parallel variant of WSABIE (*e.g.*, by averaging gradients from various machines), which might outperform RoBiRank on the MSD dataset. While performance on a specific dataset might be better, we would lose global convergence guarantees. Therefore, rather than obsess over the performance of a specific algorithm on a specific dataset, via this paper we hope to draw the attention of the community to the need for developing principled parallel algorithms for this important problem.

**Acknowledgments**   We thank anonymous reviewers for their constructive comments, and Texas Advanced Computing Center for infrastructure and support for experiments. This material is partially based upon work supported by the National Science Foundation under grant No. IIS-1117705.

## Footnotes

[1]http://sourceforge.net/p/lemur/wiki/RankLib

[2]http://research.microsoft.com/en-us/um/beijing/projects/letor/letor4dataset.aspx

[3]http://www.mcs.anl.gov/research/projects/tao/index.html

[4]the original data also provides the number of times a song was played by a user, but we ignored this in our experiment.

[5]The graph for RoBiRank 1 is hard to see because it was run for only 100,000 CPU-seconds.

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
