[Supplementary Material]

| Dataset | RoBiRank | Identity Loss | Robust Loss |
|---------|----------|---------------|-------------|
| TD 2003 | 0.9719 | 0.9704 | 0.9704 |
| TD 2004 | 0.9708 | 0.9672 | 0.9674 |
| HP 2003 | 0.9960 | 0.9947 | 0.9950 |
| HP 2004 | 0.9967 | 0.9941 | 0.9943 |
| MQ 2007 | 0.8903 | 0.8856 | 0.8783 |
| MQ 2008 | 0.9221 | 0.8857 | 0.9205 |
| MSD | 30.93% | 16.97% | 16.83% |

Table 1: Comparison of RoBiRank against Identity Loss and Robust Loss as described in Section A. We report overall NDCG for experiments on small-medium datasets, while on the Million Song Dataset (MSD) we report Precision@1.

## A   Experiments on Advantage of Using Robust Transformation

In this section, we evaluate the advantage of RoBiRank loss function (11) over simpler alternatives. The first obvious baseline is the objective function of Buffoni et al. [7], which we introduced in (6):

$$L_{\text{identity}}(\omega) := \sum_{x \in \mathcal{X}} c_x \sum_{y \in \mathcal{Y}_x} v\left(W_{xy}\right) \cdot \sum_{y' \in \mathcal{Y}_x, y' \neq y} \sigma_0\left(f_\omega(x, y) - f_\omega(x, y')\right), \qquad (18)$$

which has an advantage of being a convex objective function. While RoBiRank applies robust transformation in the following way

$$L_{\text{RoBiRank}}(\omega) := \sum_{x \in \mathcal{X}} c_x \sum_{y \in \mathcal{Y}_x} v\left(W_{xy}\right) \cdot \rho_1 \left( \sum_{y' \in \mathcal{Y}_x, y' \neq y} \sigma_0\left(f_\omega(x, y) - f_\omega(x, y')\right) \right) \qquad (19)$$

a simpler alternative would be the following:

$$L_{\text{Robust}}(\omega) := \sum_{x \in \mathcal{X}} c_x \sum_{y \in \mathcal{Y}_x} v\left(W_{xy}\right) \cdot \sum_{y' \in \mathcal{Y}_x, y' \neq y} \rho_1\left(\sigma_0\left(f_\omega(x, y) - f_\omega(x, y')\right)\right), \qquad (20)$$

which simply applies robust transformation on each of the logistic losses, instead on the sum of them as RoBiRank does. For convenience of reference, we will call (18) and (20) Identity loss and Robust loss, respectively.

We followed the same experimental protocol as in Section 3.4 and Section 4.2, and results can be seen in Table 1 and Figure 3. Identity loss and Robust loss had little difference between them; RoBiRank shows clear advantages over other baselines on TD2004, HP2004 and Million Song Dataset (MSD), and performs at least as well as others on rest of the datasets.

## B   Additional Results for Learning to Rank Experiments

In appendix A, we present results from additional experiments that could not be accommodated in the main paper due to space constraints. Figure 5 shows how RoBiRank fares against InfNorm-Push and IRPush on various datasets we used. Figure 6 shows a similar comparison against the 8 algorithms present in RankLib. Table 2 provides descriptive statistics of all the datasets we ran our experiments, Overall NDCG values obtained and values of the corresponding regularization parameters. Overall NDCG values have been omitted for the RankLib algorithms as the library doesn't support its calculation directly.

### B.1   Sensitivity to Initialization

We also investigated the sensitivity of parameter estimation to the choice of initial parameter. We initialized $\omega$ randomly with 10 different seed values. Blue lines in Figure 4 show mean and standard deviation of NDCG values at different levels of truncation; as can be seen, even though our objective function is non-convex, L-BFGS reliably converges to solutions with similar test performance. This conclusion is in line with the observation of Ding [10]. We also tried two more variants; initialization by all-zeroes (red line) and the solution of RankSVM (black line). In most cases it did not affect the quality of solution, but on TD 2003 and HP 2004 datasets, zero initialization gave slightly better results.

Figure 3: Comparison of RoBiRank with other baselines (Identity Loss and Robust Loss), see Section A

| Name | $|\mathcal{X}|$ | avg. $|\mathcal{Y}_x|$ | Mean NDCG | | | | | Regularization Parameter | | | | |
|---|---|---|---|---|---|---|---|---|---|---|---|---|
| | | | RoBiRank | RankSVM | LSRank | InfNormPush | IRPush | RoBiRank | RankSVM | LSRank | InfNormPush | IRPush |
| TD 2003 | 50 | 981 | 0.9719 | 0.9219 | 0.9721 | 0.9514 | 0.9685 | $10^{-5}$ | $10^{-3}$ | $10^{-1}$ | 1 | $10^{-4}$ |
| TD 2004 | 75 | 989 | 0.9708 | 0.9084 | 0.9648 | 0.9521 | 0.9601 | $10^{-6}$ | $10^{-1}$ | $10^{4}$ | $10^{-2}$ | $10^{-4}$ |
| Yahoo! 1 | 29,921 | 24 | 0.8921 | 0.7960 | 0.871 | 0.8692 | 0.8892 | $10^{-9}$ | $10^{3}$ | $10^{4}$ | 10 | $10^{-9}$ |
| Yahoo! 2 | 6,330 | 27 | 0.9067 | 0.8126 | 0.8624 | 0.8826 | 0.9068 | $10^{-9}$ | $10^{5}$ | $10^{4}$ | 10 | $10^{-7}$ |
| HP 2003 | 150 | 984 | 0.9960 | 0.9927 | 0.9981 | 0.9832 | 0.9939 | $10^{-3}$ | $10^{-1}$ | $10^{-4}$ | 1 | $10^{-2}$ |
| HP 2004 | 75 | 992 | 0.9967 | 0.9918 | 0.9946 | 0.9863 | 0.9949 | $10^{-4}$ | $10^{-1}$ | $10^{2}$ | $10^{-2}$ | $10^{-2}$ |
| OHSUMED | 106 | 169 | 0.8229 | 0.6626 | 0.8184 | 0.7949 | 0.8417 | $10^{-3}$ | $10^{-5}$ | $10^{4}$ | 1 | $10^{-3}$ |
| MSLR30K | 31,531 | 120 | 0.7812 | 0.5841 | 0.727 | - | - | 1 | $10^{3}$ | $10^{4}$ | - | - |
| MQ 2007 | 1,692 | 41 | 0.8903 | 0.7950 | 0.8688 | 0.8717 | 0.8810 | $10^{-9}$ | $10^{-3}$ | $10^{4}$ | 10 | $10^{-6}$ |
| MQ 2008 | 784 | 19 | 0.9221 | 0.8703 | 0.9133 | 0.8929 | 0.9052 | $10^{-5}$ | $10^{3}$ | $10^{4}$ | 10 | $10^{-5}$ |

Table 2: Descriptive Statistics of Datasets and Experimental Results in Section 3.4.

Figure 4: Performance of RoBiRank based on different initialization methods

Figure 5: Comparison of RoBiRank, RankSVM, LSRank [14], Inf-Push and IR-Push

Figure 6: Comparison of RoBiRank, MART, RankNet, RankBoost, AdaRank, CoordAscent, Lamb-daMART, ListNet and RandomForests

## C  Pseudocode of the Serial Algorithm

---

**Algorithm 1** Serial parameter estimation algorithm for latent collaborative retrieval

---
$\eta$: step size
**repeat**
  // $(U,V)$-step
  **repeat**
    Sample $(x,y)$ uniformly from $\Omega$
    Sample $y'$ uniformly from $\mathcal{Y} \setminus \{y\}$
    $U_x \leftarrow U_x - \eta \cdot \xi_{xy} \cdot \nabla_{U_x} \sigma_0(f(U_x, V_y) - f(U_x, V_{y'}))$
    $V_y \leftarrow V_y - \eta \cdot \xi_{xy} \cdot \nabla_{V_y} \sigma_0(f(U_x, V_y) - f(U_x, V_{y'}))$
  **until** convergence in $U, V$
  // $\xi$-step
  **for** $(x,y) \in \Omega$ **do**
    $\xi_{xy} \leftarrow \frac{1}{\sum_{y' \neq y} \sigma_0(f(U_x, V_y) - f(U_x, V_{y'})) + 1}$
  **end for**
**until** convergence in $U, V$ and $\xi$

---

## D  Description of Parallel Algorithm

Suppose there are $p$ number of machines. The set of contexts $\mathcal{X}$ is randomly partitioned into mutually exclusive and exhaustive subsets $\mathcal{X}^{(1)}, \mathcal{X}^{(2)}, \ldots, \mathcal{X}^{(p)}$ which are of approximately the same size. This partitioning is fixed and does not change over time. The partition on $\mathcal{X}$ induces partitions on other variables as follows: $U^{(q)} := \{U_x\}_{x \in \mathcal{X}^{(q)}}$, $\Omega^{(q)} := \{(x,y) \in \Omega : x \in \mathcal{X}^{(q)}\}$, $\xi^{(q)} := \{\xi_{xy}\}_{(x,y) \in \Omega^{(q)}}$, for $1 \leq q \leq p$.

Each machine $q$ stores variables $U^{(q)}, \xi^{(q)}$ and $\Omega^{(q)}$. Since the partition on $\mathcal{X}$ is fixed, these variables are local to each machine and are not communicated. Now we describe how to parallelize each step of the algorithm: the pseudo-code can be found in Algorithm 2.

---

**Algorithm 2** Multi-machine parameter estimation algorithm for latent collaborative retrieval

---
1: $\eta$: step size
2: **repeat**
3:   // parallel $(U,V)$-step
4:   **repeat**
5:     Sample a partition $\{\mathcal{Y}^{(1)}, \mathcal{Y}^{(2)}, \ldots, \mathcal{Y}^{(p)}\}$ **for all machine** $q \in \{1, 2, \ldots, p\}$ **do in parallel**
6:       Fetch all $V_y \in V^{(q)}$
7:       **repeat**
8:         Sample $(x,y)$ uniformly from $\{(x,y) \in \Omega^{(q)}, y \in \mathcal{Y}^{(q)}\}$
9:         Sample $y'$ uniformly from $\mathcal{Y}^{(q)} \setminus \{y\}$
10:         $U_x \leftarrow U_x - \eta \cdot \xi_{xy} \cdot \nabla_{U_x} \sigma_0(f(U_x, V_y) - f(U_x, V_{y'}))$
11:         $V_y \leftarrow V_y - \eta \cdot \xi_{xy} \cdot \nabla_{V_y} \sigma_0(f(U_x, V_y) - f(U_x, V_{y'}))$
12:       **until** predefined time limit is exceeded
13:     **end for**
14:   **until** convergence in $U, V$
15:   // parallel $\xi$-step
    **for all machine** $q \in \{1, 2, \ldots, p\}$ **do in parallel**
16:     Fetch all $V_y \in V$
17:     **for** $(x,y) \in \Omega^{(q)}$ **do**
18:       $\xi_{xy} \leftarrow \frac{1}{\sum_{y' \neq y} \sigma_0(f(U_x, V_y) - f(U_x, V_{y'})) + 1}$
19:     **end for**
20:   **end for**
21: **until** convergence in $U, V$ and $\xi$

---

$(U, V)$**-step**    At the start of each $(U, V)$-step, a new partition on $\mathcal{Y}$ is sampled to divide $\mathcal{Y}$ into $\mathcal{Y}^{(1)}, \mathcal{Y}^{(2)}, \ldots, \mathcal{Y}^{(p)}$ which are also mutually exclusive, exhaustive and of approximately the same size. The difference here is that unlike the partition on $\mathcal{X}$, a new partition on $\mathcal{Y}$ is sampled for every $(U, V)$-step. Let us define $V^{(q)} := \{V_y\}_{y \in \mathcal{Y}^{(q)}}$. After the partition on $\mathcal{Y}$ is sampled, each machine $q$ fetches $V_y$'s in $V^{(q)}$ from where it was previously stored; in the very first iteration which no previous information exists, each machine generates and initializes these parameters instead. Now let us define $L^{(q)}(U^{(q)}, V^{(q)}, \xi^{(q)}) :=$

$$\sum_{(x,y) \in \Omega^{(q)}, y \in \mathcal{Y}^{(q)}} -\log_2 \xi_{xy} + \frac{\xi_{xy} \left( \sum_{y' \in \mathcal{Y}^{(q)}, y' \neq y} \sigma_0(f(U_x, V_y) - f(U_x, V_{y'})) + 1 \right) - 1}{\log 2}.$$

In parallel setting, each machine $q$ runs stochastic gradient descent on $L^{(q)}(U^{(q)}, V^{(q)}, \xi^{(q)})$ instead of the original function $L(U, V, \xi)$. Since there is no overlap between machines on the parameters they update and the data they access, every machine can progress independently of each other. Although the algorithm takes only a fraction of data into consideration at a time, this procedure is also guaranteed to converge to a local optimum of the *original* function $L(U, V, \xi)$ according to Stratified Stochastic Gradient Descent (SSGD) scheme of Gemulla et al. [12]. The intuition is as follows: if we take expectation over the random partition on $\mathcal{Y}$, we have $\nabla_{U,V} L(U, V, \xi) =$

$$q^2 \cdot \mathbb{E} \left[ \sum_{1 \leq q \leq p} \nabla_{U,V} L^{(q)}(U^{(q)}, V^{(q)}, \xi^{(q)}) \right], \tag{21}$$

while the expectation is over the selection of the partition $\{\mathcal{Y}^{(1)}, \mathcal{Y}^{(2)}, \ldots, \mathcal{Y}^{(p)}\}$. Therefore, although there is some discrepancy between the function we take stochastic gradient on and the function we actually aim to minimize, in the long run the bias will be washed out and the algorithm will converge to a local optimum of the objective function $L(U, V, \xi)$. Specifically, (21) ensures that Condition 7 of Theorem 1 in Gemulla et al. [12] is satisfied, while the rest of conditions can be easily met by introducing an $L_2$ regularizer and thus bounding the parameter space.

$\xi$**-step**    In this step, all machines synchronize to retrieve every entry of $V$. Then, each machine can update $\xi^{(q)}$ independently of each other. When the size of $V$ is very large and cannot be fit into the main memory of a single machine, $V$ can be partitioned as in $(U, V)$-step and updates can be calculated in a round-robin way.

Note that this parallelization scheme requires each machine to allocate only $\frac{1}{p}$-fraction of memory that would be required for a single-machine execution. Therefore, in terms of space complexity the algorithm scales linearly with the number of machines.