[Reviews · NeurIPS 2014]

Submitted by Assigned_Reviewer_12

This work address the problem of learning a ranking prediction function that optimizes (N)DCG. The authors propose a surrogate loss based on a non-convex upper bound of the DCG, inspired from robust classification losses. The difference with other existing non-convex upper-bound resides in the fact that the authors introduce the non-convexity at the context level (on a whole query) and not at the pair of items level (see [8]). Then, the authors propose two applications of their algorithm with experimental studies: one is learning a prediction model for a search engine problem, the other to learn a representation for collaborative filtering.
First of all, the paper is clearly written and easy to understand even if the beginning could probably be shorten a bit (the authors’ work is only introduced p.4/8) to have space to detail more the experimental results.
The paper is based on a new non-convex surrogate loss which is a variation of a convex pairwise loss aiming at tightening the bound with the DCG.
The application of the non-convex loss proposed by the authors to the search engine setting is convincing even if the experimental results can still be improved
- No confidence intervals are provided and the figure presented in the paper is not really representative of the overall performance. I would prefer the comparison based on one of the big datasets (Yahoo / MSLR30k), maybe less impressive, but looking more representative of the performance.
On the contrary, the application to latent collaborative retrieval is more novel and is of better interest. The linearization of the non-convex loss that allows sampling over pairs of items for the optimization as well as parallelization is the main contribution. The experiments for this part are convincing especially when including the experiments that enlighten the incremental improvement w.r.t. a convex parallel baseline.
Summary: I find the proposed method of interest and I like the easy-to-follow writing of the paper. I'm convinced by the set of experiments the authors provided in answer to the reviews.

Submitted by Assigned_Reviewer_24

The paper proposes a new ranking algorithm motivated by the connection between evaluation metrics and loss function for robust classification. The performance is competitive and the authors also discuss some extension of RoBiRank to large scale data sets.

[Pros]
1. The observation of connection between evaluation metrics and loss function for robust classification.
2. The proposed algorithm is new and has been demonstrated to have good empirical performance on extensive benchmark data as compared to several state-of-the-art ranking algorithms.
3. The proposed stochastic optimization strategy is new and reasonable.

[Cons]
1. The connection of evaluation metrics and loss function is only valid when there is no ties with pairwise comparison, which should be stated clearly in the definition of (4) of the paper.
2. In the experiments, the evaluation under NDCG is also applicable. You can treat the groundtruth of items a user has listened as 1, and the ground-truth of other items as 0.
3. As the authors has pointed, the last experiment is not fair, maybe some stronger baselines should be added to the comparison.
4. It lacks some theoretical analysis of the new stochastic optimization strategy.
Summary: The authors propose a new ranking algorithm based on the theoretical findings that there is some close connection between ranking evaluation metrics and loss functions for robust classification. The technique is sound, the motivation of the paper is clear and the paper is well written. The algorithm has also been empirically proved competitive. Furthermore, the parallization of RoBiRank is interesting and important for large scale setting.

Submitted by Assigned_Reviewer_33

In this paper, the authors proposed a ranking algorithm named RoBiRank, which is motivated by their interesting observation of the connection between evaluation metrics for learning to rank and loss functions for robust classification. The proposed RoBiRank shows competitive performance comparing with recent advances, and further the authors extend the algorithm in order to be applied for large scale problems.

Quality: this paper brings the connection between two seemly unrelated problems -- robust classification and learning to rank under a general framework, and proposes to address the problem of learning to rank from a robust binary classification perspective. The proposed loss function is not convex, and the authors propose to optimize the lower bound of the metrics, which is continuous and twice differentiable. The standard l2 regularizer is applied to the model, and the optimization is performed via the standard gradient-based optimization approach. The empirical results are extensive given the use of multiple real-world datasets and comparing with 8 existing algorithms. In addition, the authors further proposed the linearization trick leading to parallelization for the optimization, and demonstrated the performance on the Million Song Dataset; by comparing with Weston et al, the authors demonstrated that their algorithm outperforms the earlier work.

Clarity: the paper is overall well written, and there are just a few very minor problems that I have mentioned below.

Originality: the proposed new loss function for ranking can be seen as (1) applying logistic loss on the difference between the item-context utility between an observed pairs, and (2) further applying a transformation function with slower rate that can lead to robustness. Note that the transformation function used in the Step 2 has been proposed in Ding [9] as cited in the paper. Given these facts and the use of the standard l2 norm on the weight, I found that the model itself is incremental. But I feel that the parallelization for the optimization algorithm is non-trivial.

Significance: the proposed model and algorithm enjoyed better performances evaluated with different metrics (e.g., precision, NDCG), based on which I think the paper is making significant contributions to the machine learning community.

I have some questions/comments below:
1. The notation \Y_{x} does not seem to be define when first introduced in line no. 119. It seems that this notation is defined later in line no. 255. and I suggest that the authors correct this minor problem.

2. Could the authors provide some analyses concerning how tight is bound for the objective function in (10)?

3. Concerning the evaluation metrics, I am wondering if the NDCG is the correct metric used to report the results. The reason is that the proposed model in this paper is optimization this metric, while for Rank-SVM the objective function is not NDCG.

4. could the author provide further analysis regarding the convergence rate for the proposed algorithm?
Summary: In this paper, the authors proposed a new model for ranking, which is motivated by observing the relation between robust binary classification and learning to rank. The main contributions consist of the new model and the scalable and parallelizable stochastic optimization algorithm. Results on a number of real-world datasets and comparing with the state-of-the-art algorithms demonstrate that the proposed model and algorithm performed better.
Author Feedback
Author rebuttal: ______________________________________________________________________________________

We thank the reviewers for their comments.

* Reviewer 1

> The application of the non-convex loss ... is close to existing
> works ([Zhang, Buffoni et al.])

Applying the \rho function to "bend" the rank is a very crucial
insight. Please see new experimental results below.

> No confidence intervals are provided

We are comparing against 12 state-of-the-art methods using 10
datasets (please see pages 10 to 14 in the supplementary
material). Sweeping over ~10 parameter values for each method
requires ~1200 jobs per fold. Doing five fold cross validation is
simply infeasible given our computational budget! However, we will
add error bars for the two largest datasets in the camera ready.

> it would have been appreciable to also have a parallel baseline
> (e.g. replacing \rho_1 by the identity result in the convex loss of
> [Zhang, Buffoni et al.] and can be parallelize the same way
> without linearization)

Thanks for pointing out this connection. This is equivalent to
Algorithm 1 (appendix B) without the \xi-step. Call this variant
Identity.

Instead of using \rho_1 on the sum of logistic losses, a related
formulation uses a robust transformation \rho_1 on the individual
logistic losses. Call this variant Robust.

We did new experiments with the two variants, and the Precision@1
values on the million song dataset are as follows:

RobiRank 29%
Identity 17%
Robust 19%

As can be seen, RobiRank comprehensively outperforms both
variants. We will include these experiments in the camera ready.

We experimented with Identity on a subset of the ranking datasets,
and in all cases RobiRank outperforms:

HP2003 HP2004 TD2003 TD2004 MQ2007 MQ2008
RobiRank 0.9960 0.9967 0.9719 0.9708 0.8903 0.9221
Identity 0.985515 0.984131 0.957544 0.945566 0.797328 0.803942

> for a empirical study, I would have like a more complete and more
> commented set of experiments.

We are comparing against 12 algorithms on 10 different
datasets. Moreover, for the parallel experiment we picked the
strongest competitor we could find, and implemented it on our own
(since open source code is not available). We will also add new
experiments with the Identity and Robust variants discussed
above. Finally, our code will be open sourced.

* Reviewer 2
> The connection of evaluation metrics and loss function is only
> valid when there is no ties with pairwise comparison, which
> should be stated clearly in the definition of (4) of the paper.

We will clarify this in the camera ready.

> You can treat the groundtruth of items a user has
> listened as 1, and the ground-truth of other items as 0.

Thanks for the suggestion. This results in a metric which is the sum
of inverse log-ranks. We will include it in the camera ready.

> As the authors has pointed, the last experiment is not fair,
> maybe some stronger baselines should be added to the comparison.

We will add two variants Identity and Robust described above as baselines.

> It lacks some theoretical analysis of the new stochastic
> optimization strategy.

Please see our response to Reviewer 3 below.

* Reviewer 3
> Note that the transformation function used in the Step 2 has been
> proposed in Ding [9] as cited in the paper.

There is a fundamental difference. We are applying the \rho_1 function
to the sum of logistic functions, while Ding et al. apply it to the
individual logistic losses i.e., \rho_1(\sum_{y \neq y'}
\sigma(f(x,y) - f(x,y'))) vs \sum_{y \neq y'} \rho_1(\sigma(f(x, y) -
f(x, y'))). This is the variant Robust discussed above, and RobiRank
significantly outperforms it.

> The notation \Y_{x} does not seem to be define when first
> introduced

This will be fixed.

> Could the authors provide some analyses concerning how tight is
> bound for the objective function in (10)?

It is hard to come up with a theoretically tight bound, because it
depends on the values of f(x, y) for all y \in \Y_{x}.

> I am wondering if the NDCG is the correct metric used to report the
> results.

For the learning to rank problem, NDCG is a widely accepted
metric. For the large scale recommendation problem we also report
Precision@k.

> could the author provide further analysis regarding the convergence
> rate for the proposed algorithm?

We can prove global convergence to a local optimum using results from
'Stochastic Approximation and Recursive Algorithms and Application' by
Kushner and Yin (2003); specifically, Theorem 6.1.1 and the discussion
in Section 6.2 applies to our algorithm. This will be included in the
supplementary material along with the final submission.

However, the rate of convergence of stochastic optimization is not
known even for the matrix completion problem, which is much simpler
than latent collaborative retrieval. Therefore, unless a significant
theoretical breakthrough is made, proving *rates* of convergence is
hard.